# Nitrogen Uptake and Translocation in *Vanda* Orchid after Roots and Leaves Application of Different Forms $^{15}$N Tracer

Kanokwan Panjama [1,2,3], Chaiartid Inkham [1,3,4], Takashi Sato [5], Takuji Ohyama [6], Norikuni Ohtake [6] and Soraya Ruamrungsri [1,2,3,*]

1    Economic Flower Crop Research Cluster, Chiang Mai University, Chiang Mai 50200, Thailand
2    Department of Plant and Soil Science, Faculty of Agriculture, Chiang Mai University, Chiang Mai 50200, Thailand
3    H.M. The King Initiative Centre for Flower and Fruit Propagation, Chiang Mai 50230, Thailand
4    Science and Technology Research Institute, Chiang Mai University, Chiang Mai 50200, Thailand
5    Faculty of Bioresource Sciences, Akita Prefectural University, Akita 010-0195, Japan
6    Gradutate School of Natural Science and Technology, Niigata University, Niigata 950-2181, Japan
*    Correspondence: sorayaruamrung@gmail.com

**Abstract:** *Vanda* is an economically important orchid that is widely produced in Thailand. Usually, growers apply large amounts of fertilizer throughout the plant, covering the leaves and roots to ensure good quality products. Nitrogen fertilizer, in terms of ammonium ($NH_4^+$) and nitrate ($NO_3^-$), is generally used as an N source. In addition, nitrogen organic fertilizer (glutamine) is increasingly being used to promote rapid growth in some plants. However, the absorption efficiency of organic N compared with the inorganic form ($NH_4^+$ and $NO_3^-$) via the roots or leaves of *Vanda* has not been evaluated. Therefore, this research aimed to compare the fate of organic N (in glutamine form) and inorganic N in *Vanda* using a $^{15}$N tracer. *Vanda* 'Patchara Delight' was grown in a plastic greenhouse under a 50% shading net at an average temperature of 25 °C and 80% relative humidity (RH). The plants were sprayed weekly via roots or leaves with 100 mL of $^{15}$N solution, 2.5 mM $^{15}$NO$_3^-$ + 2.5 mM $NH_4^+$ (N1), 2.5 mM $NO_3^-$ + 2.5 mM $^{15}$NH$_4^+$ (N2), and 2.5 mM glutamine ($^{15}$N$_2$)(N3) for 4 weeks. The plants were then sampled and separated into leaves and roots, and $^{15}$N abundance was analyzed using an elemental analyzer coupled with an isotope-ratio mass spectrometer or IRMS. The plants that received only glutamine via roots showed the highest $^{15}$N use efficiency ($^{15}$NUE) of about 28.19% at 30 days after the first feeding (DAF), whereas $^{15}$NH$_4^+$ resulted in the lowest $^{15}$NUE among $^{15}$N sources. Regardless of the application site, plants supplied with $^{15}$NH$_4^+$ showed a lower labeled N concentration and labeled N content in stems and leaves than those fed with a combination of $^{15}$NO$_3^-$ or a sole application of $^{15}$N-glutamine. The largest labeled N concentrations in stems, leaves and roots were found in plants supplied with sole glutamine via roots. At 30 DAF, $^{15}$N solution either combined with $^{15}$NO$_3^-$ or solitary $^{15}$N-glutamine did not affect the labeled N concentration in leaves. Therefore, supplying organic N in glutamine form to *Vanda* can provide a 4–7% higher NUE than inorganic N, especially when supplying the solution to the roots.

**Keywords:** glutamine; fertilizer; flower; nitrogen; accumulation

## 1. Introduction

Orchids are major flower products in Thailand, especially tropical orchids. Over 23,000 tons of orchids were exported from Thailand in 2019, valued at about 2650 million baht [1]. *Vanda* is one of the most important orchids for cut flower production in Thailand, followed by *Dendrobium* and *Phalaenopsis*, with market shares of 0.13% (cut orchid) and 8.9% (orchid plant) of the total export value [2,3]. The market potential for Thai *Vanda* is currently very desirable due to its various types of flower shapes and colors. *Vanda* is a large genus of tropical orchids. About 40 species are distributed in tropical Asia, Australia

and the Solomon Islands. It is an epiphytic orchid with leafy stems and aerial roots, but it lacks pseudobulbs [4,5].

Orchid growers generally apply liquid fertilizer to their plants by foliar application, together with fertilizer supplied directly via roots. *Vanda* is known as a heavy feeder; growers fertilize it frequently due to its lack of a pseudobulb. It is generally grown in baskets, with or without growing media. Generally, a high amount of nitrogen fertilizer is given at the vegetative stage. Subsequently, fertilizer with a high phosphorus and potassium content is applied to stimulate flowering [6]. Studies of nutrient uptake through leaves and roots have been conducted in many orchids. Hew and Yong (2004) [5] reported that 13% of $^{32}P$ was transported to the leaves when $^{32}P$-fertilizer was supplied via the roots and 19% of $^{32}P$ was found in the roots when $^{32}P$ fertilizer was supplied through the leaves. *Dendrobium* roots have N and P uptake efficiencies of 12.5 and 4.4%, respectively [5]. Ruamrunsri et al. (2014) [7] reported that *Dendrobium* Sonia 'Ear Sakul' preferred a combination of N when supplied by leaves. In *Phalaenopsis* Sogo Yukidian 'V3', nitrogen uptake efficiency was found highly in the roots followed by the leaves [8]. However, the mineral uptake efficiency of *Vanda* by foliar fertilizer and root feeding is still limited.

Although plants mostly take up inorganic forms of N, especially $NO_3^-$ and $NH_4^+$, plants can absorb organic N forms, such as glutamine. The benefit of amino acid uptake is the lower energy requirement for assimilation processes compared to absorbed inorganic N [9]. Many studies have revealed that amino acid fertilizer enhances plant growth and crop quality [10–13]. Glutamine is also an essential amino acid for plant physiology and plays a key role in the nitrogen uptake pathway and also other metabolic and biochemical reactions in plants [14–17].

The overuse of chemical fertilizer causes surface and ground water pollution, increases susceptibility to plant diseases, and decreases the number of microorganisms in the soil [18]. The weakness of Thailand's orchid production is a lack of technology and innovation to improve orchid quality, as well as a high investment in fertilizer. Thus, the aim of this study was to determine the effect of the application site and N source on N uptake and translocation in *Vanda* hybrids. If organic N is a preferential form for Vanda, especially supplying it via roots, then $^{15}N$ concentration will increase more than those received via leaves. The results of this experiment will provide beneficial information for orchid growers.

## 2. Materials and Methods

### 2.1. Plant Materials and Growth Conditions

Three-year-old *Vanda* 'Patchara Delight' plants were acclimatized by growing in the experimental location (H.M. the King Initiative Centre for Flower and Fruit Propagation, Hang Dong District, Chiang Mai, Thailand) for 3 months before starting the experiment. The plants were placed in plastic basket (roots were exposed to air) under a 50% shading house with a maximum light intensity of 905 μmol m$^{-2}$ s$^{-1}$, 33/16 °C (day/night) temperature and 80% relative humidity (RH). The plants were watered daily using deionized water, except on the day that fertilizer was applied.

### 2.2. $^{15}N$ Feeding and Analysis

Plants were supplied with two different factors, i.e., factor (1) application site including leaves or roots, and factor (2) N sources, including (N1) 2.5 mM $^{15}NO_3^-$ + 2.5 mM $NH_4^+$; (N2) 2.5 mM $NO_3^-$ + 2.5 mM $^{15}NH_4^+$; and (N3) 2.5 mM glutamine ($^{15}N_2$), in which the total amount of N was 7 mgN. All solutions contained 100 mg/L P, 200 mg/L K, 100 mg/LCa, 100 mg/LMg, 0.3 mg/LFe, 0.226 mg/LMn, 0.11 mg/LZn, 0.01 mg/LCu, 0.005 mg/LMo and 0.24 mg/LB. The experimental design was factorial in a completely randomized design (factorial in CRD) with 2 × 4 factorial combination treatments and 3 replications (plants) per treatment. $^{15}NO_3^-$ was prepared from 60 atom% of Na$^{15}NO_3$. $^{15}NH_4^+$ was derived from 60 atom% of ($^{15}NH_4$)$_2$SO$_4$, and L-glutamine ($^{15}N_2$) was derived from 98 atom% of L-glutamine ($^{15}N_2$). Plants were supplied with $^{15}N$ solution once a week for 4 consecutive weeks at 1, 8, 15 and 22 days.

Plants were sampled at 7 and 30 days after the first feeding (DAF) to investigate N uptake and translocation into various organs. Plants were separated into leaves, stems and roots and carefully washed three times with deionized water. The fresh and dry weight of plant organs were measured. To determine plant dry weight, each plant organ was dried at 80 °C for 7 days. The samples were ground into a fine powder and packed in a tin cup for total N and $^{15}$N abundance analysis. The N concentration and $^{15}$N uptake were analyzed in three replicates per treatment using an elemental analyzer (Flash EA1112; Thermo Electron, Milan, Italy) coupled with an isotope-ratio mass spectrometer or IRMS (Delta Plus XP; Thermo Fisher Scientific, Bremen, Germany).

*2.3. Calculations*

Total N distribution in each organ was calculated by following equation

$$\text{Distribution of total N content in each organ} = \frac{\text{Total N content in each organ}}{\text{Total N content in whole plant}} \times 100$$

$^{15}$N use efficiency ($^{15}$NUE) was calculated from the ratio between the amount of labeled N in the plant and the amount of labeled N supplied and expressed as a percentage [19].

$$^{15}\text{NUE} = \frac{\text{labeled N in plant}}{\text{labeled N supplied}} \times 100$$

where labeled N in the plant at each DAF equals $^{15}$N content (mg plant$^{-1}$) and labeled N supplied at each DAF is the total amount of fertilizer $^{15}$N applied. The $^{15}$N increment percentage was calculated from the $^{15}$N concentration (µg g$^{-1}$ DW) that accumulated between 7 and 30 DAF ($^{15}$N30DAF$-^{15}$N7 DAF), using the following equation:

$$^{15}\text{N increment percentage} = \frac{^{15}\text{N}_{30\,\text{DAF}} - ^{15}\text{N}_{7\,\text{DAF}}}{^{15}\text{N}_{7\,\text{DAF}}} \times 100$$

*2.4. Statistical Analysis*

The experiment was performed in a completely randomized design. The results are expressed as the mean of values measured from at least three replicates using Statistix 8 analytical software (SXW Tallahassee, FL, USA). The least significant difference at $p < 0.05$ was used to determine significant differences in growth parameters: total N content per plant, $^{15}$N concentration, absorbed $^{15}$N, and percentage $^{15}$N increment. Labeled N content and $^{15}$N use efficiency were calculated by using Tukey's test.

## 3. Results

*3.1. Total N Content Distribution (%)*

The N content in leaves, stems and roots, 7 and 30 DAF, is shown in Figure 1. A similar N distribution among organs was found between N1, N2 and N3. The distribution of total N content in plants at 7 DAF was highest in leaves (60%), followed by roots (35–40%) and stems (<1%), regardless of the N source and application site (Figure 1). At 30 DAF, the distribution of total N content in the stem gradually increased (about 7%) in all treatments (Figure 1). This suggests that, after feeding for 30 days, labeled N content was translocated into the stem in all N forms and application sites. Leaf N content was about 50–55%, and 40% was distributed in the roots. A similar result was reported by Panjama et al. (2018) [19], in which *Vanda* 'Ratchaburi Fuchs-Katsura' had an N content in the leaves of about 49%, in the roots of 47%, and little was found in the stem. *Vanda* comprises many pairs of thickened leaves, and the young leaves are considered sink organs. The old leaves and roots of *Vanda* act as storage organs [5].

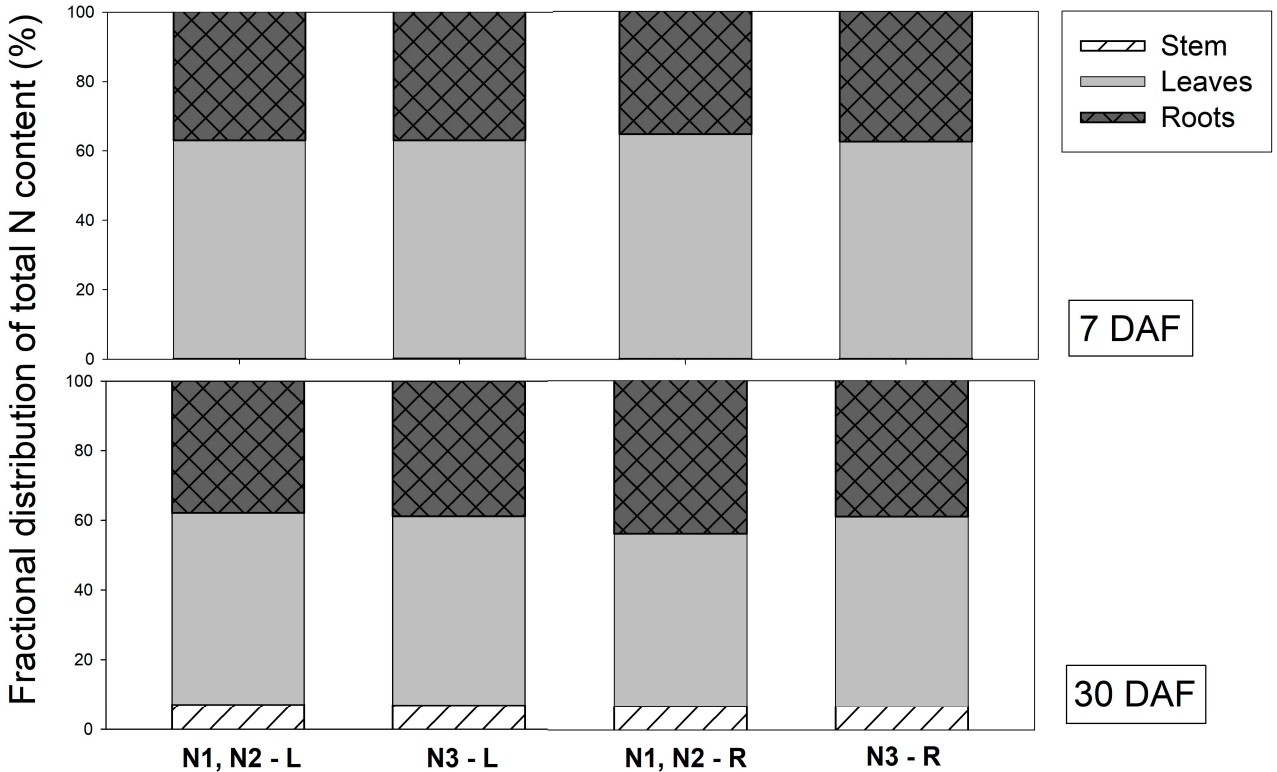

**Figure 1.** Distribution of total N in different organs of *Vanda* supplied with different N sources (N1 = 2.5 mM $^{15}NO_3^-$ + 2.5 mM $NH_4^+$, N2$^+$ = 2.5 mM $NO_3^-$ + 2.5 mM $^{15}NH_4^+$, and N3 = 2.5 mM glutamine) via leaves (L) and roots (R) at 7 and 30 days after feeding (DAF).

N was mainly distributed in the leaves at 7 and 30 DAF, although it was applied via the roots. This indicates that all N forms rapidly translocate from roots to leaves after fertilizer application.

### 3.2. Labeled N Concentration in Vanda Leaves, Stems and Roots

N applications via leaves and roots affected labeled N concentrations in *Vanda* organs (Table 1). At 7 DAF, a high concentration of labeled N was observed in the roots when plants were supplied with $^{15}N$ solution via roots, whereas the concentration was not different in the leaves and stems when the plants received $^{15}N$ solution via leaves or roots. At 30 DAF, $^{15}N$ application via roots resulted in higher labelled $^{15}N$ concentrations in stems and roots, with averages of 50.8 and 151 µgLN/gDW, respectively (Table 1). These results indicate that roots are the favored organ for mineral absorption. *Vanda* roots are covered with velamen, which have multiple spongy-like cells in the outer layer. This special, multiple-layered epidermis acts in water storage and cell protection [17,20]. However, when plants were supplied with $^{15}N$ solution via leaves, a high labeled N concentration in leaves was also observed. Many studies have reported that *Vanda* absorbs N from both leaves and roots [19,21,22]. The roots of *Vanda* played an important role as an N source after the uptake of the fertilizer solution, which was translocated to accumulate and be distributed in the leaves, as shown in Figure 1.

**Table 1.** N sources and application parts effected by the labeled N concentration in *Vanda* 'Patchara Delight' organs.

| Application Site | $^{15}$N Concentration (µgLN/gDW) at 7 DAF | | | $^{15}$N Concentration (µgLN/gDW) at 30 DAF | | |
|---|---|---|---|---|---|---|
| | **Leaves** | **Stem** | **Roots \*** | **Leaves** | **Stem \*** | **Roots \*** |
| Leaves | 6.86 | 9.60 | 36.82 b | 28.62 | 34.86 b | 75.36 b |
| Roots | 7.67 | 11.21 | 48.17 a | 32.55 | 50.78 a | 150.98 a |
| LSD ($p \leq 0.05$) | ns | ns | 7.91 | ns | 1.90 | 14.74 |
| **N sources** | **Leaves \*** | **Stem \*** | **Roots \*** | **Leaves \*** | **Stem \*** | **Roots \*** |
| 2.5 mM $^{15}$NO$_3^-$ + 2.5 mM NH$_4^+$ | 8.38 a | 10.21 ab | 36.37 b | 34.83 a | 41.80 b | 96.57 b |
| 2.5 mM NO$_3^-$ + 2.5 mM $^{15}$NH$_4^+$ | 5.48 b | 8.66 b | 30.65 b | 23.63 b | 32.33 c | 90.70 b |
| 2.5 mM Glutamine ($^{15}$N$_2$) | 7.92 a | 12.34 a | 60.48 a | 33.30 a | 54.33 a | 152.24 a |
| LSD ($p \leq 0.05$) | 1.82 | 3.35 | 9.69 | 6.76 | 2.33 | 18.05 |
| **Interaction factor** | **Leaves \*** | **Stem \*** | **Roots \*** | **Leaves \*** | **Stem \*** | **Roots \*** |
| Leaves × 2.5 mM $^{15}$NO$_3^-$ + 2.5 mM NH$_4^+$ | 8.43 a | 10.68 ab | 35.17 cd | 34.54 a | 30.01 e | 49.36 d |
| Leaves × 2.5 mM NO$_3^-$ + 2.5 mM $^{15}$NH$_4^+$ | 4.82 b | 7.49 b | 23.22 d | 19.69 b | 28.20 e | 61.93 d |
| Leaves × 2.5 mM Glutamine ($^{15}$N$_2$) | 7.32 ab | 10.63 ab | 52.09 b | 31.64 a | 46.36 c | 114.81 c |
| Roots × 2.5 mM $^{15}$NO$_3^-$ + 2.5 mM NH$_4^+$ | 8.33 a | 9.74 ab | 37.58 c | 35.13 a | 53.60 b | 143.78 b |
| Roots × 2.5 mM NO$_3^-$ + 2.5 mM $^{15}$NH$_4^+$ | 6.14 ab | 9.84 ab | 38.08 c | 27.57 ab | 36.45 d | 119.48 bc |
| Roots × 2.5 mM Glutamine ($^{15}$N$_2$) | 8.53 a | 14.06 a | 68.86 a | 34.95 a | 62.29 a | 189.67 a |
| LSD ($p \leq 0.05$) | 2.58 | 4.73 | 13.71 | 9.55 | 3.29 | 25.53 |
| CV(%) | 19.95 | 25.56 | 18.13 | 17.56 | 4.32 | 12.68 |

\* Means with the same letter within a column are not significantly different by LSD ($p < 0.05$).

Regardless of the application site, at 7 DAF, the highest labeled N concentration in all organs was obtained from $^{15}$N-glutamine (N3) supplied plants. The greatest labeled N concentration in leaves was found in plants that received $^{15}$NO$_3^-$ (N1) compared with $^{15}$NH$_4^+$ (N2), indicating that orchids plants can uptake NO$_3^-$-N better than NH$_4^+$-N. At 30 DAF, N1 and N3 treatments produced the highest labeled N concentration in leaves, while a sole $^{15}$N-glutamine application showed the highest labeled N concentration in stems and roots (54.3 and 152 µgLN/gDW, respectively; Table 1). Labeled organic N application showed a high $^{15}$N concentration in all organs. These results indicate that glutamine, as the source of organic N, is preferential for *Vanda*. Generally, plants need energy to assimilate the absorbed inorganic nitrogen to form amino acids. Compared to inorganic N, the uptake of organic N as an amino acid in plants is more advantageous in terms of the lower energy required for the ammonium assimilation metabolism [9]. In addition, the highest labeled N concentration in the leaves was also found in plants that received $^{15}$NO$_3^-$ + NH$_4^+$. It seems that *Vanda* leaves preferentially absorb N in NO$_3^-$ form. Interestingly, the labeled N concentration in all organs was lower when the plants were supplied with $^{15}$NH$_4^+$ and NO$_3^-$ (N2). NH$_4^+$ is an unfavored N form for *Vanda*. Panjama et al. (2018) [19] reported that NO$_3^-$ is the preferred form in *Vanda* 'Ratchaburi-Fusch Katsura' leaves compared to NH$_4^+$. However, when *Vanda* leaves received NH$_4^+$, it was transported to the roots and assimilated into amino acids. NH$_4^+$ can cause acidification in cells from H$^+$ release for N assimilation, which leads to ammonium toxicity symptoms in plants [23].

The combination factors between the application site and N sources altered the labeled N concentration in *Vanda* organs. Plants supplied with N$_2$ via leaves produced the lowest labeled N concentration in leaves, stems and roots at 7 and 30 DAF. According to the N source factor, NH$_4^+$ is an unfavored form of uptake in *Vanda* leaves. However, although NO$_3^-$ is more preferential in *Vanda* leaves, $^{15}$NO$_3^-$ + NH$_4^+$ application via leaves also resulted in the lowest labeled N concentration in stems and roots at 30 DAF. This result indicates that, after applying N1 to the leaves, NO$_3^-$ likely accumulates in *Vanda* leaves

rather than moving down to the stem and roots. This may be due to slow nitrate uptake rates. Näsholm et al. (2009) [24] suggested that, in many plant species, the highest uptake rates were observed in ammonium, followed by amino acids and nitrate. However, the highest labeled N concentration in the roots was observed in plants that received $^{15}$N-glutamine via roots at 7 and 30 DAF (Table 2). This suggests that nitrogen in an organic form, such as glutamine, was highly accumulated in *Vanda* roots instead of inorganic N.

**Table 2.** N source and application site effect on labelled N content in *Vanda* 'Patchara Delight' organs.

| Application Site | $^{15}$N Content (µg$^{15}$N/Plant) at 7 DAF | | | $^{15}$N Content (µg$^{15}$N/Plant) at 30 DAF | | |
|---|---|---|---|---|---|---|
| | Leaves * | Stem * | Roots * | Leaves * | Stem * | Roots * |
| Leaves | 101.71 a | 18.23 a | 374.34 b | 444.44 a | 78.76 b | 847.80 b |
| Roots | 118.04 a | 21.25 a | 501.60 a | 508.95 a | 112.33 a | 1869.60 a |
| LSD ($p \leq 0.05$) | 23.10 | 5.55 | 92.94 | 81.14 | 8.96 | 166.31 |
| **N sources** | Leaves * | Stem * | Roots * | Leaves * | Stem * | Roots * |
| 2.5 mM $^{15}$NO$_3^-$ + 2.5 mM NH$_4^+$ | 125.25 a | 20.11 ab | 385.03 b | 522.32 a | 95.29 b | 1191.00 b |
| 2.5 mM NO$_3^-$ + 2.5 mM $^{15}$NH$_4^+$ | 84.94 b | 16.10 b | 311.68 b | 378.78 b | 73.89 c | 1156.30 b |
| 2.5 mM Glutamine ($^{15}$N$_2$) | 119.44 a | 23.01 a | 617.21 a | 528.99 a | 117.45 a | 1728.90 a |
| LSD ($p \leq 0.05$) | 28.29 | 6.80 | 113.83 | 99.37 | 10.97 | 203.69 |
| **Interaction factor** | Leaves * | Stem * | Roots * | Leaves * | Stem * | Roots * |
| Leaves × 2.5 mM $^{15}$NO$_3^-$ + 2.5 mM NH$_4^+$ | 122.18 a | 22.12 ab | 375.94 bc | 497.86 a | 73.28 de | 555.30 d |
| Leaves × 2.5 mM NO$_3^-$ + 2.5 mM $^{15}$NH$_4^+$ | 73.92 b | 12.84 b | 235.48 c | 315.72 b | 60.60 e | 680.00 d |
| Leaves × 2.5 mM Glutamine ($^{15}$N$_2$) | 109.04 ab | 19.75 ab | 511.60 b | 519.76 a | 102.39 bc | 1308.30 c |
| Roots × 2.5 mM $^{15}$NO$_3^-$ + 2.5 mM NH$_4^+$ | 128.32 a | 18.10 ab | 394.11 bc | 546.78 a | 117.30 ab | 1826.70 b |
| Roots × 2.5 mM NO$_3^-$ + 2.5 mM $^{15}$NH$_4^+$ | 95.96 ab | 19.37 ab | 387.88 bc | 441.84 ab | 87.17 cd | 1632.70 b |
| Roots ×2.5 mM Glutamine ($^{15}$N$_2$) | 129.84 a | 26.27 a | 722.82 a | 538.23 a | 132.51 a | 2149.50 a |
| LSD ($p \leq 0.05$) | 40.00 | 9.62 | 160.98 | 140.53 | 15.52 | 288.06 |
| CV (%) | 20.47 | 27.39 | 20.66 | 16.57 | 9.13 | 11.92 |

* Means with the same letter within a column are not significantly different by LSD ($p < 0.05$).

### 3.3. N Content and NUE

The labeled N content was calculated using the $^{15}$N concentration multiplied by the dry weight of each plant part. Therefore, this value presented the accumulation, assimilation and partition of N in each *Vanda* organ, and it was affected by $^{15}$N application site. Applying $^{15}$N solution via roots obviously showed a higher labeled N content in roots than those in which it was received via leaves at 7 and 30 DAF (Figure 2). The labeled N content in the leaves was not different when plants were supplied with $^{15}$N solution via leaves or roots. Our results revealed that the greatest total labeled N content in plants was observed when plants were supplied $^{15}$N solution via roots (641 and 2490 µg$^{15}$N/plant) at 7 and 30 DAF, respectively (Table 2). These results were similar to labeled N concentration data, showing that roots are a major organ absorbing $^{15}$N solution.

Regarding N sources, at 7 DAF, the labeled N content in leaves and stems was not different when plants were sprayed with N1 and N3. The lowest labeled N content in these organs was obtained from plants that received N2 when NH$_4^+$ was labelled. It was confirmed that *Vanda* did not regularly accumulate NH$_4^+$ in leaves, but rather in roots. In contrast, $^{15}$glutamine supplied plants showed the greatest labeled N content in roots (617 µg$^{15}$N/plant) and total labeled N content in plants (760 µg$^{15}$N/plant) (Figure 2). At 30 DAF, N1 and N3 had the highest labeled content in leaves, whereas the highest labeled N content in stems and roots was found in plants that received N3. The application of labeled organic N also resulted in the greatest total labeled N content per plant of about 2380 µg$^{15}$N/plant (Table 2).

The labeled N content in *Vanda* organs varied among treatments. Interestingly, $^{15}$glutamine application via roots showed the greatest amount of labeled N content in all organs at 7 and 30 DAF. This treatment also resulted in the highest total labeled N content per plant, at 879 and 2820 $\mu g^{15}N$/plant at 7 and 30 DAF, respectively. In addition to this treatment, plants sprayed with $^{15}NO_3^- + NH_4^+$ via leaves had the greatest labeled N content in leaves at 7 and 30 DAF.

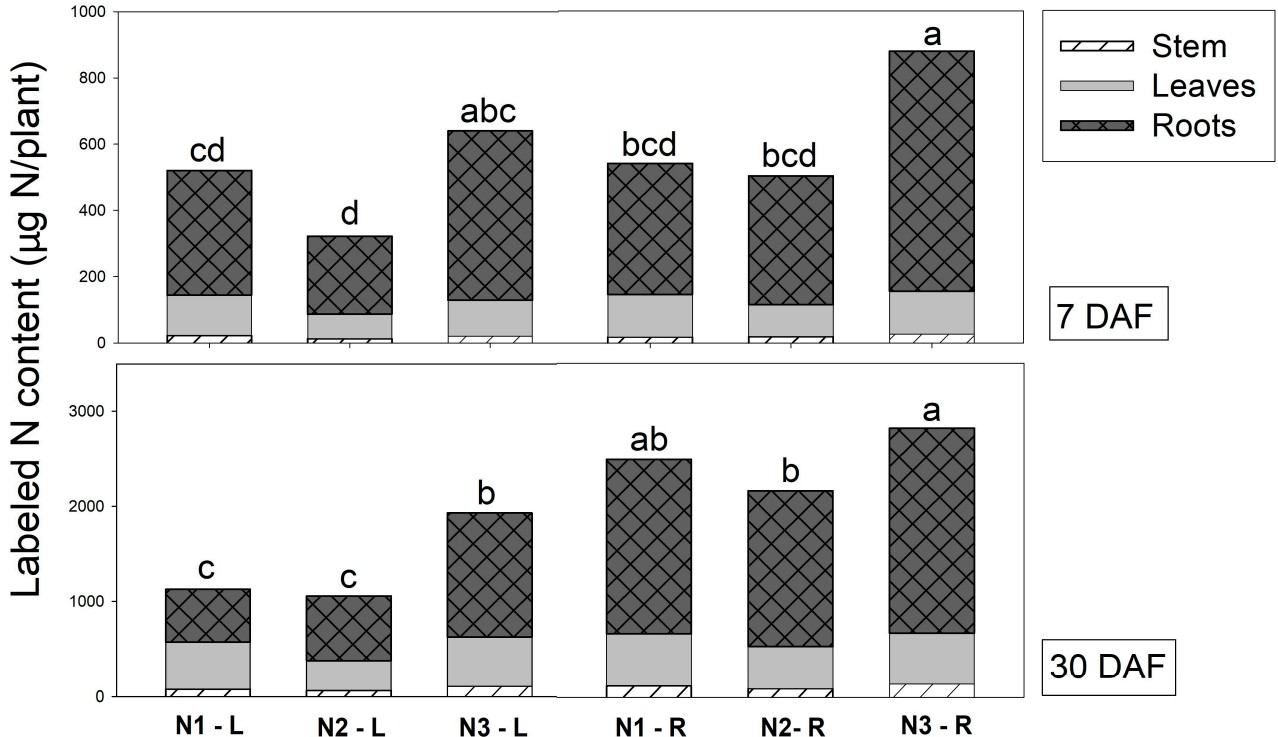

**Figure 2.** Total N content ($\mu g^{15}N$/plant) in different organs of *Vanda* supplied with different labeled N sources (N1 = 2.5 mM $^{15}NO_3^- + 2.5$ mM $NH_4^+$, N2 = 2.5 mM $NO_3^- + 2.5$ mM $^{15}NH_4^+$, and N3 = 2.5 mM glutamine) via leaves (L) and roots (R) at 7 and 30 days after feeding (DAF). The different letters denote significant differences in treatments, analyzed by Tukey's test.

Regarding N sources, $^{15}$N-glutamine obviously showed an NUE value higher than inorganic N treatments. It was found that $^{15}$N-glutamine application via roots had an NUE higher than other treatments at 25.1 and 28.2, on average, at 7 and 30 DAF (Figure 3). Glutamine, as an N organic form, produced a higher NUE than inorganic N forms, especially when it was supplied via roots. Our results indicate that organic N is preferentially absorbed by roots rather than leaves. Similar results were also found in another epiphyte orchid, Phalaenopsis, where the roots are a major organ absorbed nitrogen more efficiency and readily translocated it when compared to application via leaves [8]. The exogenous application of amino acids has been shown to induce growth promotion effects on many crops [11,14,15,25,26]. Han et al. (2022) [27] reported that external application of L-glutamine enhanced NUE in poplar 'Nanlin895' higher than that of inorganic N.

The $^{15}$N increment percentage in *Vanda* was calculated. Plants supplied with $^{15}$N solution via roots had a higher $^{15}$N increment than those supplied via leaves. When plants were sprayed with $^{15}NO_3^- + NH_4^+$ via roots, a 350% increase was observed (Figure 4). The lowest of $^{15}$N increment percentage was obtained from $^{15}NO_3^- + NH_4^+$ via leaf treatment, with a value of about 120%. The preferential absorbing site was confirmed by the increment percentage value (Figure 4). Regardless of the N source, the increment percentage of labeled N in plants receiving the solution via roots was higher than in those fed via leaves. This clearly indicates that roots are a major organ for mineral absorption in *Vanda*. Similar results was found in Phalaenopsis, their roots preferentially uptake nutrition rather than

the leaves [8]. Wang (2010) [28] reported that only foliar fertilization was not sufficient for Phalaenopsis good growth, but supplement fertilizer through leaves and roots enhanced the growth and flowering of the plant. Thus, we conclude that fertilizer N should mainly be applied to the roots of Vanda.

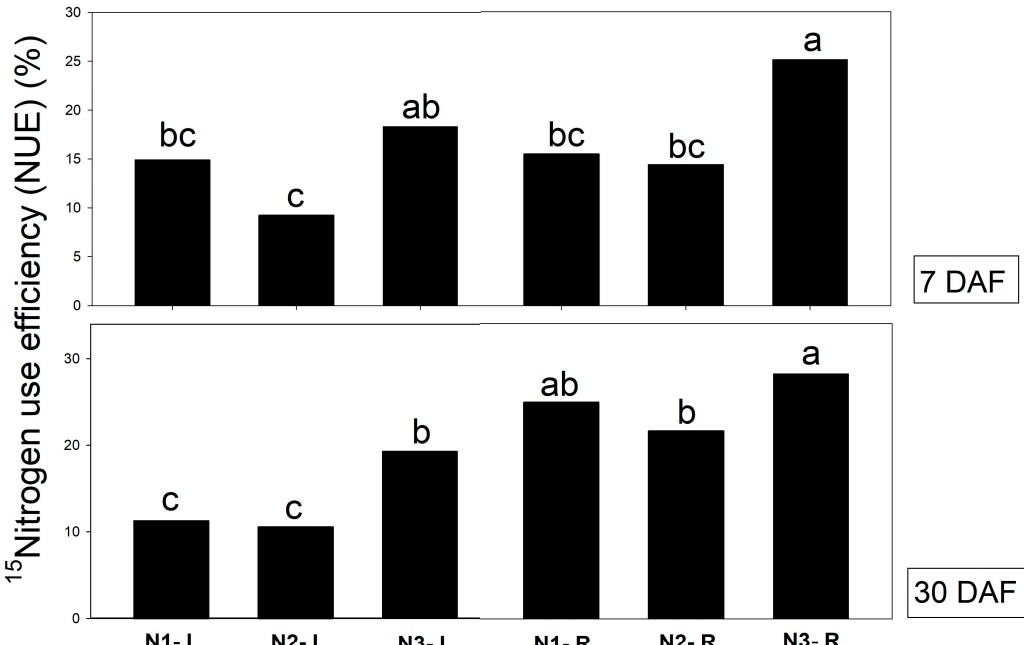

**Figure 3.** NUE in different organs of *Vanda* supplied with different labeled N sources (N1 = 2.5 mM $^{15}NO_3^- + 2.5$ mM $NH_4^+$, N2 = 2.5 mM $NO_3^- + 2.5$ mM $^{15}NH_4^+$, and N3 = 2.5 mM glutamine) via leaves (L) and roots (R) at 7 and 30 days after feeding (DAF). The different letters denote significant differences in treatments, analyzed by Tukey's test.

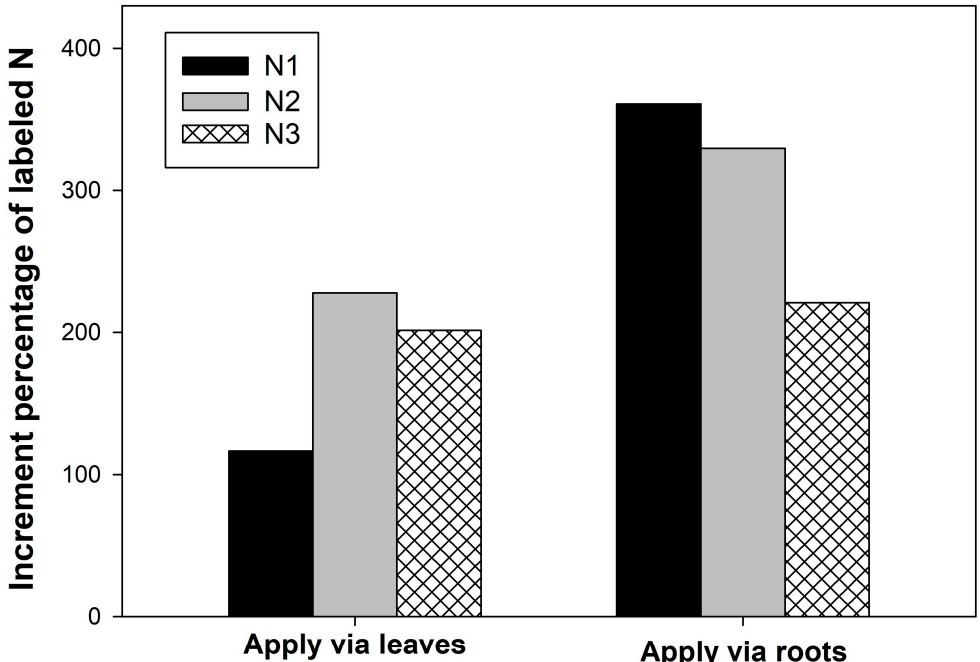

**Figure 4.** Increment percentage of labeled N in *Vanda* supplied with different N sources via leaves and roots. (N1 = 2.5 mM $^{15}NO_3^- + 2.5$ mM $NH_4^+$, N2 = 2.5 mM $NO_3^- + 2.5$ mM $^{15}NH_4^+$, and N3 = 2.5 mM glutamine).

### 3.4. Distribution of Labeled N Content in Vanda Leaves, Stems and Roots

The distribution of labeled N content in *Vanda* organs was observed. At 7 DAF, the labeled N was mainly distributed in the roots of plants in all treatments, which was nearly 75% of the labeled N distribution, and the remaining N was mainly present in the leaves and the stem (Figure 5). Plants fed with glutamine, regardless of the application site, had a percentage of labeled N in roots of about 80%, which was higher than other N sources. At 30 DAF, the labeled N distribution in the leaves increased when the plants were supplied with labeled nitrate combined and ammonium via leaves (45%). However, application via roots showed an equal distribution of labeled N in organs among all $^{15}$N solutions, 20% in leaves, 5% in the stem, and 75% in roots. This indicates that *Vanda* preferred mineral absorption via the roots. *Vanda* roots comprise the velamen, sponge-like layered epidermis cells that naturally store water and plant nutrition. Generally, plants uptake water and mineral elements from roots and translocate them upward to the shoot via the xylem using the transpiration stream, capillary force or mass flow. In contrast with *Dendrobium*, leaves are major organs that absorb plant nutrition [7]. Interestingly, when $^{15}$N was applied via leaves, about 70% of labeled N was assimilated and translocated downward to the roots via the phloem, and only 20% was found in the leaves. After fertilizer was applied to the *Vanda* leaves, nitrogen rapidly moved down to the roots. At 30 DAF, the distribution of labeled N in leaves gradually increased to 40% when the plant was fed with $^{15}$NO$_3^-$ + NH$_4^+$ via the leaves, demonstrating that nitrate is the preferred N form for uptake in *Vanda* leaves. However, when the plants were fed with both inorganic and organic N forms via the roots, about 70% of the labeled N accumulated in the roots and 20% was translocated to the leaves, followed by 5% in the stem. The distribution percentage of labeled N changed slightly at 7 DAF, suggesting that roots are a major organ for absorbed minerals in *Vanda*.

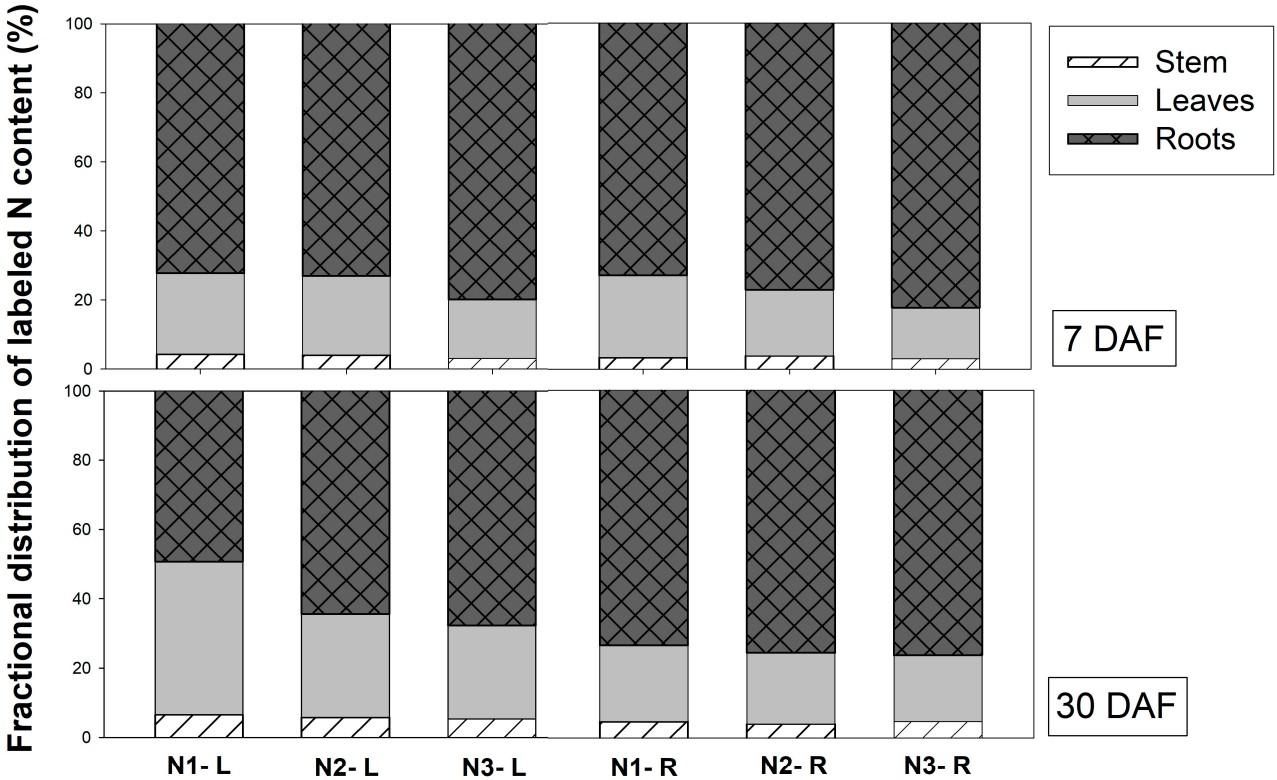

**Figure 5.** Distribution of labeled N in different organs of *Vanda* supplied with different labeled N sources (N1 = 2.5 mM $^{15}$NO$_3^-$ + 2.5 mM NH$_4^+$, N2 = 2.5 mM NO$_3^-$ + 2.5 mM $^{15}$NH$_4^+$, and N3 = 2.5 mM glutamine) via leaves (L) and roots (R) at 7 and 30 days after feeding (DAF).

## 4. Conclusions

Labeled N concentrations in stems and roots were significantly increased when $^{15}$N fertilizer was supplied via roots compared with foliar application. $^{15}$Glutamine application resulted in a higher labeled N concentration and labeled N content in all organs, regardless of the application site. NUE was high when the plant was supplied with $^{15}$N-glutamine, especially via root application. Thus, from our results, supplying organic fertilizer as glutamine directly via the roots is recommend for Vanda production.

**Author Contributions:** Conceptualization, K.P. and S.R.; methodology, K.P., C.I. and S.R.; software, K.P.; validation, K.P. and S.R.; formal analysis, K.P., C.I. and S.R.; investigation, K.P., C.I. and S.R.; data curation, T.O., T.S. and S.R.; writing—original draft preparation, K.P., C.I. and S.R.; writing—review and editing, K.P. and S.R.; visualization, K.P.; supervision, T.S., T.O., N.O. and S.R.; project administration, K.P. and S.R.; funding acquisition, K.P. and S.R. All authors have read and agreed to the published version of the manuscript.

**Funding:** This research received no external funding.

**Data Availability Statement:** K.P. and S.R. are responsible for data keeping, and data are available upon request.

**Acknowledgments:** This research was partially supported by Chiang Mai University. We thank H.M. and the King's Initiative Centre for Flower and Fruit Propagation, Chiang Mai, Thailand, for their kind support.

**Conflicts of Interest:** The authors declare no conflict of interest.

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
