# Peer review of "Nitrogen Uptake and Translocation in Vanda Orchid after Roots and Leaves Application of Different Forms 15N Tracer"

_horticulturae, doi:10.3390/horticulturae8100902_

Round 1

Reviewer 1 Report

Dear Authors

In the manuscript entitled "Amino acid uptake and translocation in Vanda Orchid via roots and leaves using 15N tracer", ID Horticulturae-1915992, the authors present a comparison of various methods of nitrogen fertilization on the content (concentration), uptake (accumulation), and distribution of this macronutrient in the tested plant. Before publication, the manuscript requires revision. I present my suggestions below.

Title

I suggest modifying it by the authors.

My proposal: "Nitrogen uptake and translocation in Vanda Orchid after roots and leafs application of different forms 15N tracer".

Abstract

I propose to shorten the introduction to the abstract a bit (lines 15-22). In the following, important information, indicated by the subject of the manuscript, i.e. nitrogen translocation depending on the site and form of its application, is missing.

1. Introduction

Lines 56-62: Solid literature support is required, not just one item

Lines 70-71: Literature Reference is required

An aim of the research: the information about nitrogen translocation in the tested plant should be included

Research hypothesis: it would be good to formulate the research hypothesis

2. Materials and Methods

Subsection 2.1: Plant materials and growth conditions

There is no detailed information about the place and conditions under which the experiment was conducted (eg vegetation hall, containers / vases, etc.?). The description is too short for the reader.

Subsection 2.2: 15N feeding and analysis

Lines 87-94: I propose to use SI-compatible units.

It is unfortunate the authors did not take into account the fourth fertilizer treatment with urea solution, also enriched with the 15N isotope. This organic form of nitrogen is often used for foliar fertilization of plants and would be perfect for comparison with the organic form of nitrogen when applied as glutamine.

Subsection 2.3: Calculations

Lines 87-94: calculation formula, line 109

I propose to write: "labeled 15N in plant" and "labeled 15N supplied"

Please explain why "15N increment percentage" was not calculated as 15N increment between 0 DAF and 30 DAF. Before the application of fertilizers with 15 N, the enrichment of the plant with the 15N isotope can be taken as a zero value, the so-called "Day zero". After the application of the first dose of fertilizer on the first day and sampling on the seventh day, the plants had already taken up and contained nitrogen from this fertilizer, including some enrichment in 15N. Why was this nitrogen pool omitted from the 15N increment percentage calculation?

Subsection 2.4: Statistical analysis

Line 118: How many repetitions were there? Please enter some value.

3. Results and Discussion

Subsection 3.1: Total N distribution (%)

Lines 125-133: Figure 1 included in supplementary file does not provide cited data (treatments N1, N2 and N3). Figure 1 requires correction.

Please clarify: does this subsection contain a description of the distribution of total N calculated from "nitrogen content" or "nitrogen uptake"? The description shows the proportion of nitrogen accumulation. Please make it clear.

Subsection 3.2: Labeled N concentration in Vanda leaves, stems and roots

Line 141: Please indicate specific data that shows this fact.

Lines 146-148: In these sentences there is a description that is mutually exclusive.

Lines 161-162: Please complete the missing marks with the same letters (as explained).

Figure 1. Please provide in subsection 2.3 how the distribution of total N in different organs of Vanda was calculated.

Applies to all Figures and Tables captions: I propose to replace detailed description of fertilizer treatments with abbreviations explained in the test methodology - N1, N2, and N3.

In subsection 3.3 and Figure 2 shows N content or N uptake (accumulation)? Please explain.

Overall, the discussion of results is quite short. In the discussion of the results, one should refer to the impact of a better nitrogen supply of plant on, for example, the operational parameters of the Vanda Orchid.

4. Conclusions

In chapter 1, in lines 75-76 it says: "The results of this experiment will provide beneficial information for orchid growers." Unfortunately, no practical recommendations are given in the summary.

Overall comments

I suggest major revision of the manuscript.

Yours faithfully

Reviewer

Author Response

Dear Authors

In the manuscript entitled "Amino acid uptake and translocation in Vanda Orchid via roots and leaves using 15N tracer", ID Horticulturae-1915992, the authors present a comparison of various methods of nitrogen fertilization on the content (concentration), uptake (accumulation), and distribution of this macronutrient in the tested plant. Before publication, the manuscript requires revision. I present my suggestions below.

Response to reviewer: Thank you very much for your kind consideration this manuscript as well as your valuable comments that help to improve this manuscript.

Title

I suggest modifying it by the authors.

My proposal: "Nitrogen uptake and translocation in Vanda Orchid after roots and leafs application of different forms 15N tracer".

Response to reviewer: Thank you for your kind suggestion. We have changed the title as your suggestion.

Abstract

I propose to shorten the introduction to the abstract a bit (lines 15-22). In the following, important information, indicated by the subject of the manuscript, i.e. nitrogen translocation depending on the site and form of its application, is missing.

Response to reviewer: Thank you for your valuable advice. The abstract was shortened, and some missing data were added (Lines 14-22 with yellow highlight).

  1. Introduction

Lines 56-62: Solid literature support is required, not just one item

Response to reviewer: Thank you for your comment, further literature was added [Lines 63-66 with yellow highlight].

Lines 70-71: Literature Reference is required

Response to reviewer: The literature reference was added [Line 15 with yellow highlight].

An aim of the research: the information about nitrogen translocation in the tested plant should be included

Response to reviewer: Thank you for your suggestion, we have rewritten the aim of the research [Lines 78]

Research hypothesis: it would be good to formulate the research hypothesis

Response to reviewer: Thank you for a significant advice, we have formulated the hypothesis of this research in Lines 79-80 with yellow highlight.

  1. Materials and Methods

Subsection 2.1: Plant materials and growth conditions

There is no detailed information about the place and conditions under which the experiment was conducted (eg vegetation hall, containers / vases, etc.?). The description is too short for the reader.

Response to reviewer: We have added location and growing condition as well as growing method followed your suggestion [Lines 86 – 91 with yellow highlight].

Subsection 2.2: 15N feeding and analysis

Lines 87-94: I propose to use SI-compatible units.

Response to reviewer: We have rewritten the unit in SI as your suggestion [Lines 99-101 with yellowing highlight].

It is unfortunate the authors did not take into account the fourth fertilizer treatment with urea solution, also enriched with the 15N isotope. This organic form of nitrogen is often used for foliar fertilization of plants and would be perfect for comparison with the organic form of nitrogen when applied as glutamine.

Response to reviewer: Thank you for the invaluable advice. We agree with you about conducting experiments in urea, and we will continue to experiment in the future.

Subsection 2.3: Calculations

Lines 87-94: calculation formula, line 109

I propose to write: "labeled 15N in plant" and "labeled 15N supplied"

Response to reviewer: We have edited as your suggestion, line 120 with yellow highlight.

Please explain why "15N increment percentage" was not calculated as 15N increment between 0 DAF and 30 DAF. Before the application of fertilizers with 15 N, the enrichment of the plant with the 15N isotope can be taken as a zero value, the so-called "Day zero". After the application of the first dose of fertilizer on the first day and sampling on the seventh day, the plants had already taken up and contained nitrogen from this fertilizer, including some enrichment in 15N. Why was this nitrogen pool omitted from the 15N increment percentage calculation?

Response to reviewer: Because of the formula of 15N increment percentage when compare between 0 DAF and 30 DAF = [(15N 30 DAF – 15N 0 DAF)/ 15N 0 DAF] × 100 which, when representing 15N at 0 DAF is Zero, causes the resulting value to be equal to 0, so we start to calculate the value at 7 DAF.

 As for a reason we chose to start the analysis on 7 DAF, because we think that the nitrogen absorbed by plants is likely to accumulate to a greater amount. Bring to a clearer picture of distribution than the analysis at day 1 DAF.    

Subsection 2.4: Statistical analysis

Line 118: How many repetitions were there? Please enter some value.

Response to reviewer: The experimental design, treatments and replications were added and indicated in Lines 101-103 with yellow highlight.

  1. Results and Discussion

Subsection 3.1: Total N distribution (%)

Lines 125-133: Figure 1 included in supplementary file does not provide cited data (treatments N1, N2 and N3). Figure 1 requires correction.

Response to reviewer: Thank you for your comments, we have corrected Figure  1 and provided cited data as your suggestion.

Please clarify: does this subsection contain a description of the distribution of total N calculated from "nitrogen content" or "nitrogen uptake"? The description shows the proportion of nitrogen accumulation. Please make it clear.

Response to reviewer: We have changed Subsection 3.1: Total N distribution (%) to “Total N content distribution (%)” [Line 137 , 139, 141, 143 with yellow highlight].

Subsection 3.2: Labeled N concentration in Vanda leaves, stems and roots

Line 141: Please indicate specific data that shows this fact.

Response to reviewer: We decided to delete this sentence in revised manuscript “The labeled N data showed the N concentration derived from only the fertilizer solution” [Line 153 with yellow highlight].

Lines 146-148: In these sentences there is a description that is mutually exclusive.

Response to reviewer: We have deleted the sentence [Lines 159 -160 with yellow highlight]

to avoid mutually exclusive.

Lines 161-162: Please complete the missing marks with the same letters (as explained).

Response to reviewer: We have completed the marks in both table 1 and 2 with yellow highlight.

Figure 1. Please provide in subsection 2.3 how the distribution of total N in different organs of Vanda was calculated.

Response to reviewer: The equation to calculate the distribution of total N in each organ was added in subsection 2.3 [Lines 116-117 with yellow highlight].

Applies to all Figures and Tables captions: I propose to replace detailed description of fertilizer treatments with abbreviations explained in the test methodology - N1, N2, and N3.

Response to reviewer: Thank you for your comments, all figures and tables were replaced detailed description as your suggestion.

In subsection 3.3 and Figure 2 shows N content or N uptake (accumulation)? Please explain.

Response to reviewer: Figure 2 shows N content because it was calculated from plant dry weight.

Overall, the discussion of results is quite short. In the discussion of the results, one should refer to the impact of a better nitrogen supply of plant on, for example, the operational parameters of the Vanda Orchid.

Response to reviewer: Thank you for your suggestion, we have provided additional discussion in Lines 226-229 , 240 – 244 with yellow highlight.

  1. Conclusions

In chapter 1, in lines 75-76 it says: "The results of this experiment will provide beneficial information for orchid growers." Unfortunately, no practical recommendations are given in the summary.

Response to reviewer: Thank you for your comment, we have added practical recommendations in the summary [Lines 273 -275 with yellow highlight].

Overall comments

I suggest major revision of the manuscript.

Response to reviewer: Thank you for your kind consideration our manuscript as major revision as well as your valuable comments.

Thank for reviewer for the thoughtful and thorough review. Hopefully we have addressed all your concerns.

Sincerely Yours,
Soraya Ruamrungsri

Reviewer 2 Report

Dear authors,

this is a very well presented and written study, showing interesting and important results.

Minor typos, errors, and formatting requirements are highlighted in the manuscript.

Line 167: orchid plants

Line 246: This clearly indicates that roots are a major organ for mineral absorption in Vanda. How about other orchids, or other taxa? Please elaborate further. Compare those results with general knowledge. 

Lines 256-259: same comment as previous. Add references regarding the statement in line 259.

The conclusion section should follow the investigation goal and provide major advantages and applicability. 

Since the aim was that 'the results of this experiment provide beneficial information for orchid growers' please clearly state them in the conclusions. 

Which fertilizers are a recommendation following your results? How many spraying/treatments? What are the main benefits of such an approach? 

Author Response

Dear authors,

This is a very well presented and written study, showing interesting and important results.

Minor typos, errors, and formatting requirements are highlighted in the manuscript.

Response to reviewer: Thank you very much for your kind consideration our manuscript.

Line 167: orchid plants

Response to reviewer: We have rewritten in this sentence [Line 170]

Line 246: This clearly indicates that roots are a major organ for mineral absorption in Vanda. How about other orchids, or other taxa? Please elaborate further. Compare those results with general knowledge.

Response to reviewer: Thank you for your suggestion, we have added the result in other orchid in Line 240 -244 with yellow highlight.

Lines 256-259: same comment as previous. Add references regarding the statement in line 259.

Response to reviewer: The reference was added as your suggestion [Line 258 -259].

The conclusion section should follow the investigation goal and provide major advantages and applicability.

Response to reviewer: The practical recommendation was added in the conclusion [Line 273-275].

Since the aim was that 'the results of this experiment provide beneficial information for orchid growers' please clearly state them in the conclusions.

Response to reviewer: The practical recommendation was added in the conclusion [Line 273-275].

Which fertilizers are a recommendation following your results? How many spraying/treatments? What are the main benefits of such an approach?

Response to reviewer: From our results, apply glutamine 2.5 mM via roots of Vanda once a week  is recommended for Vanda production, since it’s showed the highest of nitrogen use efficiency.

Thank for reviewer for the thoughtful and thorough review. Hopefully we have addressed all your concerns.

Sincerely Yours,

Prof. Dr. Soraya Ruamrungsri

Reviewer 3 Report

Thanks for the opportunity to review the interesting manuscript.

Here I would like to highlight my comments and suggestions,

(1) In this paper, 15N isotope tracing technique was used to compare the effects of orchids on the absorption, transfer and distribution of inorganic and organic nitrogen, which has certain theoretical value. Compared with ornamental plants, it is more significant for rational fertilization and green production to study the transfer and distribution of nitrogen in grains for food crops. In this sense, the research lacks urgency in actual production.

(2) One concern that I have is that the reported work was done in pot experiment with one variety. Only one cultivar was used. Since we know that there are genotypic differences in responses to so many things, shouldn't the authors comment on the need for evaluation of other genotypes, and the likelihood that there will be cultivator differences in responses?

(3) The experiment design did not explain clearly. What are the test factors and their levels?

Moreover, the implementation steps of the test are too simple.

Lines 87-88 “The 15N tracer was fed to plants once via leaves or roots with 100 ml of different 87 labeled N sources” How did the author spray 15N on the leaves? How to ensure no nitrogen loss?

Lines 92-93 “15NO3- was prepared from 60 atom% of Na15NO3. 15NH4+ was derived from 60 atom% 92 of (15NH4)2SO4, and L-glutamine (15N2) was derived from 98 atom% of L-glutamine (15N2).” Whether the nitrogen content of three different forms of 15N is the same?

(4) The conclusions part should be revised thoroughly. The results need to be summarized and then draw a conclusion.

Generally, I think this manuscript is not suitable for publication in this journal.

Author Response

Thanks for the opportunity to review the interesting manuscript.

Here I would like to highlight my comments and suggestions,

(1) In this paper, 15N isotope tracing technique was used to compare the effects of orchids on the absorption, transfer and distribution of inorganic and organic nitrogen, which has certain theoretical value. Compared with ornamental plants, it is more significant for rational fertilization and green production to study the transfer and distribution of nitrogen in grains for food crops. In this sense, the research lacks urgency in actual production.

 Response to reviewer: Thank you very much for the valuable comments. Since Thailand is rank number one of tropical orchid exporter in the world, orchid growers supplied a large amount of chemical fertilizer to ensure good quality of orchids. Chemical fertilizer costed a high investment in Thailand orchid production. According to our previous studied [1] showing that NUE is depends on N sources and little amount of N fertilizer in chemical forms (Nitrate and Ammonium) was uptake in Vanda. Our hypothesis is if N in amino acid form is rapidly and readily absorbed by Vanda plant then application amino acid directly may improve N uptake in plant.

(2) One concern that I have is that the reported work was done in pot experiment with one variety. Only one cultivar was used. Since we know that there are genotypic differences in responses to so many things, shouldn't the authors comment on the need for evaluation of other genotypes, and the likelihood that there will be cultivator differences in responses?

 Response to reviewer: Thank you for your comments. We agree with you about concerning other genotypic responses. However, this research was aimed and focus on Vanda which is an economic orchid crop in Thailand and also in the world market, the other crops are out of our interest. It is the basic research to determine the uptake and translocation of organic N and inorganic N in Vanda. Of course, the further experiment should be done in more detail including the effects of varieties and the application of fertilizer use as your concern.

(3) The experiment design did not explain clearly. What are the test factors and their levels?

Moreover, the implementation steps of the test are too simple.

 Response to reviewer: Thank you for your comments, we have added the experimental designed including factors, their levels, treatment and replication number in Line 95-103 with yellow highlight. According to methodology, analysis 15N abundance in material plant by using an elemental analyzer coupled with an isotope-ratio mass spectrometer is highly accurate and the implementation steps were adapted from previous studies [1], [2], [3]

Lines 87-88 “The 15N tracer was fed to plants once via leaves or roots with 100 ml of different 87 labeled N sources” How did the author spray 15N on the leaves? How to ensure no nitrogen loss?

 Response to reviewer:

We carefully and gently supplied the solution by foggy sprayer to leaves or roots according to previous studies [1] [3]. To prevent 15N contamination between leaves and roots, either leaves or roots were covered with plastic bag during spray the solution to other sites. The result showed how much N could be up taken in different N-forms from fertilizer (usually orchid grower apply excess N fertilizer to plant). Therefore, this result will be beneficial and useful for orchid growers to concern fertilizer application for Vanda.

Lines 92-93 “15NO3- was prepared from 60 atom% of Na15NO3. 15NH4+ was derived from 60 atom% 92 of (15NH4)2SO4, and L-glutamine (15N2) was derived from 98 atom% of L-glutamine (15N2).” Whether the nitrogen content of three different forms of 15N is the same?

 Response to reviewer: Yes, we calculated based on total amount of N which is equal in each form of 15N (7 mgN each).

(4) The conclusions part should be revised thoroughly. The results need to be summarized and then draw a conclusion.

 Response to reviewer: Thank you for your advice, we have revised the conclusion part.

Generally, I think this manuscript is not suitable for publication in this journal.

Response to reviewer: Thank for reviewer for the thoughtful and thorough review. Hopefully we have addressed all your concerns and we are really do hope that you will reconsidering our manuscript to be publication in this journal.

Sincerely Yours,

Prof. Dr. Soraya Ruamrungsri

Ref.

[1] Panjama, K.; Ohyama, T.; Ohtake, N.; Sato, T.; Potapohn, N.; Sueyoshi, K.; Ruamrungsri, S. Identifying N sources that affect N uptake and assimilation in Vanda hybrid using 15N tracers. Hortic. Environ. Biotechnol. 2018, 59, 805–813.

[2] Susilo, H.; Peng, Y.; Lee, S.; Chen, Y.; Alex Chang, Y. The uptake and partitioning of nitrogen in Phalaenopsis Sogo Yukidian ‘V3’ as shown by 15N as a tracer. J. Amer. Soc. Hort. Sci. 2013, 138(3), 229-237.

[3] Ruamrungsri, S.; Khuankaew, T.; Sato, T.; Ohyama, T. Nitrogen sources and its uptake in dendrobium orchid by 15N tracer study. Acta Hortic. 2014, 1025. 10.17660/ActaHortic.2014.1025.30.

Round 2

Reviewer 1 Report

Dear Authors

I have read the corrections introduced by the Authors and the answers to my suggestions / doubts. In my opinion, the manuscript 'ID horticulturae-1915992' entitled "Nitrogen Uptake and Translocation in Vanda Orchid After Roots and Leaves Application of Different Forms 15N Tracer" in its current version may be published in the Horticulturae journal.

Kind Regards

Reviewer

Reviewer 3 Report

I believe the manuscript has been  sufficiently improved to warrant publication in Horticulturae